# Antifungal Substances Produced by *Xenorhabdus bovienii* and Its Inhibition Mechanism against *Fusarium solani*

**DOI:** 10.3390/ijms23169040

**Published:** 2022-08-12

**Authors:** Yu Wang, Fengyu Zhang, Cen Wang, Peifeng Guo, Yeqing Han, Yingting Zhang, Bingjiao Sun, Shaojie Shan, Weibin Ruan, Jiao Pan

**Affiliations:** 1College of Life Sciences, Nankai University, Tianjin 300071, China; 2Institute for Cultural Heritage and History of Science & Technology, University of Science and Technology Beijing, Beijing 100083, China

**Keywords:** nematode endosymbiotic bacteria, cultural heritage disease fungus, antifungal substances, metabolomic analysis, GC–MS, transcriptome sequencing, inhibition mechanism

## Abstract

Fungal colonization can severely damage artifacts. Nematode endosymbiotic bacteria exhibit good prospects in protecting artifacts from fungal damage. We previously found that supernatant from the fermentation of nematode endosymbiotic bacterium, *Xenorhabdus bovienii*, is effective in inhibiting the growth of *Fusarium solani* NK-NH1, the major disease fungus in the Nanhai No.1 Shipwreck. Further experiments proved that *X. bovienii* produces volatile organic compounds (VOCs) that inhibit NK-NH1. Here, using metabolomic analysis, GC–MS, and transcriptomic analysis, we explored the antifungal substances and VOCs produced by *X. bovienii* and investigated the mechanism underlying its inhibitory effect against NK-NH1. We show that *X. bovienii* produces several metabolites, mainly lipids and lipid-like molecules, organic acids and derivatives, and organoheterocyclic compounds. The VOCs produced by *X. bovienii* showed two specific absorption peaks, and based on the library ratio results, these were predicted to be of 2-pentanone, 3-(phenylmethylene) and 1-hexen-3-one, 5-methyl-1-phenyl. The inhibition of *F. solani* by VOCs resulted in upregulation of genes related to ribosome, ribosome biogenesis, and the oxidative phosphorylation and downregulation of many genes associated with cell cycle, meiosis, DNA replication, and autophagy. These results are significant for understanding the inhibitory mechanisms employed by nematode endosymbiotic bacteria and should serve as reference in the protection of artifacts.

## 1. Introduction

Nematodes are an evolutionarily lower group of animals that can be categorized according to their food tropism as saprophytic, plant parasitic, and animal parasitic nematodes. Nematodes, which have insects as their obligate hosts, are known as entomopathogenic nematodes [1,2]. Entomopathogenic nematodes have a wide host range of up to 3000 species [3]. Entomopathogenic nematode endosymbiotic bacteria are Gram-negative bacteria belonging to Enterobacteriaceae that parasitize the gut of entomopathogenic nematodes [4].

The potential of entomopathogenic nematodes, mostly belonging to the genera *Steinernemati* sp. and *Heterorhabditis* sp., in controlling pests and disease-causing microorganisms has been focused upon since the first reports of insect parasitic nematodes in 1929 [5,6]. Bacteria, in symbiotic association with *Steinernemati* sp., belonging to the genus *Xenorhabdus* include several species, such as *X. nematophilus*, *X. poinarii*, *X. bovienii*, *X. beddingii*, and *X. japonica*. Insecticidal substances produced by *X. nematophilus* are frequently reported and have been more thoroughly studied [7]. *Photorhabdus* sp. is a symbiont of *Heterorhabditis* sp., which produces insecticidal substances, such as insecticidal proteins, proteases, lipases, and lipopolysaccharides, among others [8]. Nonvolatile metabolites have crucial roles in insecticidal killing and in inhibiting fungal and bacterial growth [9,10,11,12]; for example, the fermentation broth of nematode endosymbiotic bacteria showed inhibitory effects against various plant disease causing fungi, such as *Fusarium*, *Penicillium*, and *Rhizobia* [13,14]. Upon isolation and extraction from the fermentation broth, nonvolatile secondary metabolites, exhibiting bacteriostatic activity, were mainly found to be indoles [15], alkenes [16], and ketones [17]. Similarly, the inhibition of bacteria and fungi by nematode volatile organic compounds (VOCs), has been studied by several researchers. For example, Fernando et al. isolated two aldehydes (nonanal and n-decanal) and two alcohols (cyclohexanol and 2-ethyl-1-hexanol) from volatile gases, which were determined to be disulfide and thiazole compounds based on the presence of benzene ring [18].

Artifacts of great importance produced by human activities, historically lying in rivers, are of great importance in tracing the history and from the perspective of art and scientific research [19,20]. A wide variety of microorganisms are involved in the biocorrosion of artifacts—hyphae of fungi and actinomycetes can invade and damage the material of artifacts and bacteria can secrete a variety of secondary metabolites that corrode the artifacts [21,22,23,24]. Such biological diseases, besides affecting the preservation of artifacts, can also damage the appreciation value of artifacts. The spread of a large number of fungal hyphae and spores on the surface of artifacts obscures their original attributes [25,26]. In addition, microorganisms secrete pigment-like substances that damage the aesthetic value of artifacts. *Fusarium solani* NK-NH1 is a dominant surface disease fungus on the hull of the Nanhai No. 1 Shipwreck, which has strong cellulose and lignin degradation capacity and is largely harmful to the hull [27].

Nematode endosymbiotic bacteria are a new class of bioresources with great potential. Research on entomopathogenic nematode endosymbiotic bacteria can provide highly effective biofungicides for the control of fungi affecting artifacts. Although these bacteria have been applied to control plant disease fungi, there are no reports regarding their inhibitory effect on artifactual disease fungi. The utilization of metabolites from nematode endosymbiotic bacteria to inhibit artifactual fungal diseases, especially the utilization of bacteriostatic VOCs produced by these bacteria, has great prospects for application in cultural preservation. Previous screening in our laboratory showed that the fermentation broth of *X. bovienii* has good inhibitory effect against artifactual disease fungi. In this study, we analyzed the effective inhibitory components produced by nematode endosymbiotic bacteria using metabolomics and GC–MS and established a theoretical foundation for research on the mechanism of inhibition employed by the identified components using transcriptome sequencing with the aim of devising new strategies for the protection of artifacts.

## 2. Results

### 2.1. Antifungal Substances in the Supernatant of X. bovienii Ferment

#### 2.1.1. Inhibitory Effect of Incubation with the Supernatant of Fermented *X. bovienii* for Different Time

Previous experimental results showed that the supernatant of *X. bovienii* ferment contains secondary metabolites that inhibit the growth of fungi that deteriorate cultural heritage artifacts. To detect the antifungal substances in the supernatant of *X. bovienii* ferment, it is necessary to obtain high concentrations of secondary metabolites as test samples. The supernatant of *X. bovienii* ferment showed the maximum inhibitory effect on NK-NH1 when incubated for 2 days (Figure 1). Therefore, an incubation time of 2 days was chosen for metabolomics analysis.

#### 2.1.2. Correlation Analysis of Metabolomic Data

The results of sample correlation analysis showed that the experimental and control groups were clustered separately, and the parallelism of the samples was good (Figure 2a). Principal components analysis (PCA) analysis showed that there was no intersection between the samples in the experimental and blank groups, which indicated that the main components of the supernatant in the experimental and blank groups were different, and the samples in the experimental group clustered closely together without discrete points generated, indicating that the experimental samples were parallel and the data were reliable (Figure 2b).

#### 2.1.3. Metabolites in the Supernatant of *X. bovienii* Ferment

Antifungal substances produced by *X. bovienii* are expected to be present in the experimental group and to be absent or present in low amounts in the blank group. Therefore, we first performed Venn analysis to find metabolites unique to the experimental group. A total of 476 unique cationic metabolites and 544 unique anionic metabolites were found in the experimental group; among these there were 44 named cationic metabolites and 70 named anionic metabolites (Appendix A). These 114 named metabolites were classified using the HMDB 4.0 database. Lipids and lipid-like molecules accounted for the highest (52.75%) proportion of these metabolites, followed by organic acids and derivatives and organoheterocyclic compounds, which accounted for 16.48% and 8.79%, respectively. In addition, there were few benzenoids, organic oxygen compounds, organoxygen compounds, phenylpropanoids and polyketides, organic nitrogen compounds, and alkaloids and their derivatives (Figure 3).

Further, we screened the differential metabolites that showed a large difference in content. Using the selected screening criteria (fold change (FC) > 100; *p* value < 0.001), a total of 14 substances were screened (Table 1).

### 2.2. Volatile Antifungal Substances Produced by X. bovienii

#### 2.2.1. Inhibitory Effect of VOCs Produced by *X. bovienii* on NK-NH1

To avoid direct contact between antifungal substances and cultural relics, which may damage the relics, we searched for volatile antifungal substances. We examined whether the VOCs produced by *X. bovienii* had an inhibitory effect on fungal growth. As shown in Figure 4, in the blank group, NK-NH1 grew vigorously and the hyphae spread to the other side, whereas in the experimental group, the NK-NH1 colonies were small, the hyphae were underdeveloped, and the fungal growth was significantly inhibited. These results proved that *X. bovienii* could produce VOCs that have an inhibitory effect on NK-NH1.

#### 2.2.2. GC–MS of VOCs Produced by *X. bovienii*

The VOCs produced by *X. bovienii* were passed through a column and the individual components were separated using chromatography. The data were analyzed using software. Two peaks at 35.629 and 37.685 min in the experimental group were obviously different from those in the blank, indicating that these VOCs were produced by *X. bovienii* and might be the compounds that inhibited the growth of NK-NH1 (Figure 5). The molecular weights of these two compounds were determined to be 174 and 188, respectively, using mass spectrometry (Figure 6). Based on the results of comparison with the NIST library, these two compounds are suspected to be 2-pentanone, 3-(phenylmethyl) and 1-hexen-3-one, 5-methyl-1-phenyl.

### 2.3. Transcriptomic Analysis of NK-NH1

#### 2.3.1. Differential Expression Analysis

Using transcriptome sequencing of NK-NH1, we explored the inhibitory mechanism of VOCs produced by *X. bovienii* at the gene transcription level. We found 6788 differentially expressed genes, which included 3545 upregulated and 3243 downregulated genes in the experimental group (Figure 7).

#### 2.3.2. KEGG Enrichment Analysis of Differentially Expressed Genes

By performing the enrichment analysis of the differential gene sets, it is possible to find the biological functions or pathways that are significantly associated with the differential genes under different conditions. KEGG is a comprehensive database integrating genomic, chemical, and systematic functional information. KEGG top 20 pathways enrichment analysis showed that the differentially expressed genes were mainly associated with ribosome, ribosome biogenesis in eukaryotes, oxidative phosphorylation, autophagy, ubiquitin mediated proteolysis, peroxisome, and cell cycle (Figure 8). Many genes in the fgr0301 pathway associated with ribosome, in the fgr03008 pathway associated with ribosome biogenesis in eukaryotes, and in the fgr00190 pathway associated with oxidative phosphorylation were significantly upregulated, whereas many genes in the fgr04111 pathway associated with the cell cycle, in the fgr04113 pathway associated with meiosis, in the fgr03030 pathway associated with DNA replication, and in the fgr04138 pathway associated with autophagy were significantly downregulated (Appendix A).

## 3. Discussion

At present, considering their similarities with artifact disease fungi, plant disease fungi are mostly used in research on secondary metabolites produced in the fermentation broth of endosymbiotic bacteria. The secondary metabolites produced by nematode endosymbiotic bacteria can also be used to inhibit the growth of artifact disease fungi, and are expected to be useful in the protection of artifacts. We isolated four nematode endosymbiotic bacteria from multiple nematode strains, namely *X. nematophila*, *X. cabanillasii*, *X. bovienii*, and *P. luminescens*. In previous experiments, it was found that the fermentation products of *X. bovienii* have the best inhibitory effect on NK-NH1, a major disease fungus in the Nanhai NO. 1 Shipwreck, and the results of untargeted metabolomic analysis revealed that *X. bovienii* produced several metabolites, mainly lipids and lipid-like molecules, organic acids and derivatives, and organoheterocyclic compounds; however, only a small number of these metabolites have been identified. The antifungal components in fermentation broth have been analyzed in only a few studies. In some studies, it was only confirmed that the fermentation broth contained large amounts of substances, such as ketones, aldehydes, and phenyl rings, and no further research was conducted on specific substances [28]. Studies on secondary metabolites produced by microorganisms mostly used chemical extraction methods, and crude extracts obtained from the isolates were used in bacteriostatic tests. Crude extracts with good bacteriostatic effects were screened and then developed as bacteriostatic substances [13]. Although no specific metabolites were identified in our study, we screened 14 known compounds, mostly lipids and lipid-like molecules, produced by *X. bovienii* that with showed significant differences.

Among the numerous microbial secondary metabolites, VOCs have received increasing attention over the past decade. Since the early reports on bacterial VOCs contributing to plant health and growth, an increasing number of studies have demonstrated great potential for the application of these gaseous molecules in bacteriostasis [29]. Microbial VOCs are usually released dynamically in a variety of forms and are mainly derived from catabolic activities; they mostly comprise low complexity, lipophilic compounds [30,31]. In this study, we show that *X. bovienii* produces VOCs and effectively inhibits the growth of *F. solani*, NK-NH1. In several studies, it has been shown that VOCs inhibit the growth and development of several fungi. For example, Fernando et al. tested volatile compounds from endosymbiotic bacteria isolated from rapeseed and soybean plants for their inhibitory effects on *Sclerotinia sclerotiorum*, a phytopathogenic fungus, and found that sclerotial and hyphal growth, as well as spore germination, were inhibited [18]. Shan et al. demonstrated the growth inhibitory effect of volatile substances produced by the endosymbiotic bacteria of a nematode strain against plant disease fungi, *Mucor* sp. and *Rhizomucor* sp. [32]. GC-MS analysis revealed that the major VOCs produced by *X. bovienii* were esters, ethers, heterocycles, and ketones, and there were two metabolites that were different from those in the control group. The results of database alignment showed that these two metabolites might be 2-pentanone, 3-(phenylmethyl) and 1-hexen-3-one, and 5-methyl-1-phenyl. At present, there are few studies on the VOCs produced by nematode endosymbiotic bacteria, and most researchers focus on analyzing the effective bacteriostatic active components in their fermentation broth. Fernando et al. isolated the volatiles with better fungicidal activity using GC-MS, and identified them as nonanal, n-decanal, cyclohexanol, and 2-ethyl, 1-hexanol [18]. Shan et al. detected the main bacteriostatic component produced by nematode endosymbiotic bacteria as dimethyl disulfide using GC-MS [32].

To explore the inhibitory mechanism of VOCs produced by *X. bovienii*, in this study, we used transcriptome sequencing to analyze the changes occurring at the gene expression level upon the inhibition of NK-NH1. The expression of 6788 genes of NK-NH1 was found to change after the inhibition. Many of the genes involved in cell cycle, meiosis, and DNA replication were downregulated, which indicates the decreased proliferation of NK-NH1 after inhibition. Autophagy-related genes were also downregulated, but three genes in this metabolic pathway, KRAS (a GTPase), Snf1-activating kinase 1, and serine palmitoyltransferase were upregulated. These three genes are upstream of the autophagy regulatory pathway and encode key enzymes that are decisive in the occurrence of autophagy; their upregulation indicates autophagy may occur to some extent in cells. Autophagy in *Saccharomyces cerevisiae* is regulated by many factors, among which the most thoroughly studied are the protein kinase target of rapamycin (TOR) and the Ras/Camp-dependent protein kinase A (PKA) signal cascade system [33]. Serine palmitoyltransfer is related to TOR complex 1 (TORC1), TORC1 contains Tor1 or Tor2, as well as Kog1, Tco89, and Lst8, which can be inhibited by the immunosuppressant rapamycin and participate in the regulation of cell cycle, ribosome biogenesis, ribosome translation activity, amino acid utilization, and other growth and metabolic processes [34]. Studies have shown that TOR can sense the level of intracellular glutamine, which is the main intermediate of nitrogen metabolism and can be used as an indicator of intracellular nutritional status. Therefore, *S. cerevisiae* cells can regulate the degree of autophagy at different nutritional levels through TOR pathway to ensure the normal physiological condition of cells [35]. When cells are under nutritional stress, such as nitrogen starvation, the TOR pathway is inhibited, and autophagy can be negatively regulated by TORC1. Since the main downstream effectors of Tor are type-2A protein phosphatase (PP2A) and members of the PP2A-like protein family, they are involved in the phosphorylation of a variety of intracellular proteins [36]. When TOR is inhibited, TORC1 stimulates PP2A- and PP2A-like protein family members, dephosphorylates autophagy-related proteins Atg1 and Atg13, and can bind with Atg17 to form the Atg1-13-17 complex, thereby inducing autophagy [37]. In *S. cerevisiae*, the Ras/cAMP-dependent PKA pathway is involved in regulating cell metabolism and growth at different carbon source levels. At present, studies have shown that the Ras/cAMP-dependent PKA pathway negatively regulates autophagy. Under nitrogen starvation conditions, autophagy formation is blocked when PKA catalytic subunit Tpk1 is overexpressed in Saccharomyces cerevisiae and autophagy is strongly inhibited [38]. In addition, in vitro and in vivo experiments confirmed that Atg1 contains two potential PKA phosphorylation sites, and PKA can directly modify Atg1. The PKA phosphorylation of Atg1 leads to changes in its localization. After autophagy induction, Atg1 in the wild type can be transferred from cytoplasm to PAS site, but Atg1 mutants lacking PKA site are always located at PAS related structures [39]. These data suggest that the Ras/cAMP-dependent PKA signaling pathway inhibits autophagy by negatively regulating the binding of Atg1 to PAS. Autophagy is a self-protective mechanism in biological cells to obtain nutrients by dissolving cellular contents, reducing unnecessary life activities and maintaining only the basic metabolism. Thus, a generally hostile environment easily leads an organism to mount an autophagic response. We observed that the growth of *Fusarium* on bipartitioned plates was severely inhibited and the mycelial growth was very slow due to the action of organic volatiles produced by the endosymbiotic bacterium. Thus, the upregulation of key genes of the autophagy pathway may be required for self-protection by *Fusarium* upon the sensing of the external environment that is not conducive for growth.

## 4. Materials and Methods

### 4.1. Strains

The nematode endosymbiotic bacterium, *X. bovienii*, used in the experiment was isolated from the nematode strain, SF SN. The fungus used in the experiment was *F. solani* (NK-NH1, KY410238), which had been isolated from the hull of the Nanhai No. 1 Shipwreck and is the most dominant disease fungus on the hull of this shipwreck.

### 4.2. Inhibitory Effect of the Supernatant of X. bovienii Ferment on F. solani

The supernatant of *X. bovienii* ferment was obtained from the liquid shake culture of the bacterium grown in TSY medium, with shaking at 180 rpm at 28 °C for 1, 2, 3, 4, or 5 days. One milliliter of the supernatant from culture incubated for different time periods was added to 20 mL of PDA medium. An 8 mm block was punched out from the edge of a culture plate of *F. solani* (NK-NH1) using a hole puncher and added to the center of the plate. The experimental group was set up in triplicate. In the control group, TSY medium was added to the PDA medium and incubated in a 28 °C incubator for 5 days.

### 4.3. Metabolome Analysis

#### 4.3.1. Sample Preparation

The bacterial culture fermented for 2 days was used as the experimental group (NKSF) and TSY liquid medium was used as the control group (Blank); eight parallel cultures were set for each group. Hundred-microliter sample was taken in a 1.5 mL centrifuge tube and 400 μL extraction solution (acetonitrile:methanol = 1:1) was added to it. After vortexing the tube for 30 s, low-temperature ultrasonic extraction (5 °C, 40 KHz) was performed for 30 min. The sample was held undisturbed at −20 °C, for 30 min, and then centrifuged (13,000× *g* for 15 min) at 4 °C. The supernatant was transferred to a new tibe, blow-dried with nitrogen, and redissolved in 120 µL of a complex solution (acetonitrile:water = 1:1). The mixture was subjected to low-temperature ultrasonic extraction (5 °C, 40 KHz) for 5 min, and then centrifuged (13,000× *g* for 5 min) at 4 °C. The supernatant was transferred to an injection vial with internal cannula for machine analysis.

#### 4.3.2. Chromatography Conditions

The instrument used for LC–MS analysis was an ultra-high performance liquid chromatography tandem time of flight mass spectrometry UPLC-Triple TOF system from AB SCIEX. The sample (10 µL) was injected into a BEH C18 column (100 mm × 2.1 mm i.d., 1.8 µm) and the eluting compounds were detected after separation using mass spectrometry. The mobile phases used for separation were as follows: mobile phase A: water (containing 0.1% formic acid); mobile phase B: acetonitrile/isopropanol (1/1; containing 0.1% formic acid). The solvent gradient used for separation was as follows: 0–3 min: mobile phase A decreased linearly from 95% to 80% and mobile phase B increased linearly from 5% to 20%; 3–9 min: mobile phase A decreased linearly from 80% to 5% and mobile phase B increased linearly from 20% to 95%; 9–13 min: mobile phase A was maintained at 5% and mobile phase B was maintained at 95%; 13.0–13.1 min, mobile phase A increased linearly from 5% to 95% and mobile phase B decreased linearly from 95% to 5%; 13.1–16 min: mobile phase A was maintained at 95% and mobile phase B was maintained at 5%. The flow rate was 0.40 mL/min and the column temperature was set at 40 °C.

#### 4.3.3. Conditions Used for Mass Spectrometry

The mass spectrum signals from the samples were collected in the positive and negative ion scanning mode over a mass scanning range (m/z) of 50–1000. The parameters used for mass spectrometry were as follows: positive ion voltage, 5000 V; negative ion voltage, 4000 V; declustering voltage, 80 V; spray gas, 50 psi; auxiliary heating gas, 50 psi; air curtain gas, 30 psi, ion source heating temperature, 500 °C; cyclic collision energy, 20–60 V.

#### 4.3.4. Data Analysis

The LC–MS raw data were imported into the metabolomics processing software, Progenesis QI (Waters Corporation, Milford, CT, USA), for processing. The MS and MSMS data were analyzed using the public database HMDB (http://www.hmdb.ca/) (accessed on 25 May 2022) and Metlin (https://metlin.scripps.edu/) (accessed on 25 May 2022). The metabolites were identified based on database matching. The preprocessed data were uploaded to the Meiji Biological Cloud Platform (https://cloud.majorbio.com) (accessed on 15 June 2022) for analysis.

### 4.4. Inhibitory Effect of VOCs Produced by X. bovienii on F. solani

A two-partition plate was used for this experiment. The partition in the middle was impermeable to substances, such as liquid; however, a gap between the top portion of the partition wall and the plate cover allowed volatile gases to pass across the partition wall. Melted TSA medium was poured on one side of the partition and melted PDA medium was poured on the other side. After the solidification of the media, 50 μL of the symbiotic bacterial solution of nematode was coated on the side with the TSA medium. An 8 mm block from the edge of a tested *F. solani* (NK-NH1) colony was punched out with a hole puncher and added to the center of the side with the PDA medium. In the control group, the partition plate was not coated with the bacterial solution, and only a fungal block was placed on the side with the PDA medium. The plates were placed in an incubator set at 28 °C for 5 days.

### 4.5. Detection of VOCs Using GC-MS

#### 4.5.1. Collection of VOCs Produced by *X. bovienii*

*X. bovienii* was inoculated in 10 mL TSY agar medium in a 100 mL conical flask and incubated at 28 °C for 2 days. The VOCs were collected using solid-phase microextraction (SPME). SPME fibers (Supelco, Bellefonte, PA, USA) were pretreated in the GC inlet at 280 °C for 10 min. Thereafter, a preheated SPME fiber was inserted into the flask through the sealing film on the conical flask and exposed to the top space of the conical flask for 60 min to collect VOCs. In the control group, the conical flask only contained TSA medium and was not inoculated with bacteria. Three experimental groups and a control group were set up.

#### 4.5.2. GC–MS Detection

An HP-5 nonpolar chromatographic column (Agilent, Santa Clara, CA, USA, 30 m × 0.25 mm, 0.25 μM thin layer) was used for the separation of VOCs. The sample was manually injected in a non-shunting mode; helium was used as the carrier gas at a flow rate of 1.0 mL/min. The SPME fiber at the GC (Agilent 7890) injection port was heated to 250 °C for 5 min to realize thermal desorption. The temperature program was set as follows: 40 °C for 5 min, 4 °C for 1 min, continuous temperature rise to 100 °C, continuous temperature rise to 100 °C for 3 min, 5 °C for 1 min, continuous temperature rise to 200 °C, continuous temperature rise to 200 °C for 3 min, 5 °C for 1 min, and continuous temperature rise to 280 °C for 15 min. Mass spectrometry was performed on an Agilent 5975c system. The compounds were identified by comparing the obtained data with the known data in the NIST library.

### 4.6. Transcriptome Sequencing Analysis

NK-NH1, with significantly inhibited mycelial growth (NKXb), and the control group (CK) were used for transcriptome sequencing. The samples were sent to Beijing Novogene Co., Ltd. (Beijing, China). for analysis. Raw data can be found at https://www.ncbi.nlm.nih.gov/geo/query/acc.cgi?acc=GSE207994 (accessed on 27 June 2022).

#### 4.6.1. Differential Expression Analysis

The analysis of the differentially expressed genes in the two groups was performed using the DESeq2 R package (Version 1.16.1; Creator: Michael Love, Simon Anders, Wolfgang Huber; Location: Billerica, MA, USA). DESeq2 provides statistical routines for determining differential expression in digital gene expression data using a model based on negative binomial distribution. The resulting *p*-values were adjusted using the Benjamini and Hochberg’s approach for controlling the false discovery rate. Genes with an adjusted *p*-value < 0.05 found using DESeq2 were considered to be differentially expressed.

#### 4.6.2. KEGG Enrichment Analysis of Differentially Expressed Genes

KEGG is a database resource for understanding the high-level functions and utilities of the biological system, such as cells, organisms, and ecosystems, from molecular-level information, especially large-scale molecular datasets generated in genome sequencing and other high-throughput experimental technologies (http://www.genome.jp/kegg/ (accessed on 2 July 2022)). We used the cluster Profiler R package to test the statistical enrichment of differentially expressed genes in the KEGG pathways.

## 5. Conclusions

*X. bovienii* can produce a variety of metabolites, mainly lipids and lipid-like molecules, organic acids and derivatives, and organoheterocyclic compounds. The VOCs produced by *X. bovienii* produced two specific absorption peaks, and the library ratio results were 2-pentanone, 3-(phenylmethylene) and 1-hexen-3-one, 5-methyl-1-phenyl. The treatment of *F. solani* with VOCs resulted in the significant upregulation of genes related to ribosome, ribosome biogenesis, and oxidative phosphorylation, and the significant downregulation of genes associated with cell cycle, meiosis, DNA replication, and autophagy.

## Figures and Tables

**Figure 1 ijms-23-09040-f001:**
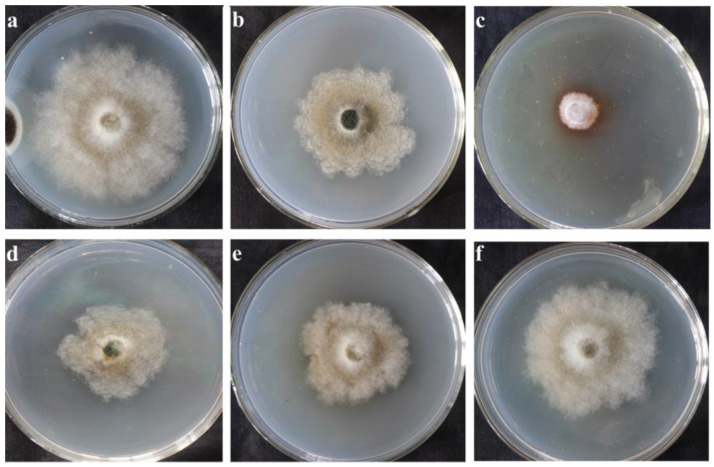
Inhibitory effect of the supernatant of *Xenorhabdus bovienii* on *Fusarium solani* (NK-NH1) incubated for different time. Incubated at 28 °C for 5 days, *n* = 3. (**a**) TSY medium; (**b**) 1-day incubation time; (**c**) 2 day incubation time; (**d**) 3-day incubation time; (**e**) 4-day incubation time; (**f**) 5-day incubation time.

**Figure 2 ijms-23-09040-f002:**
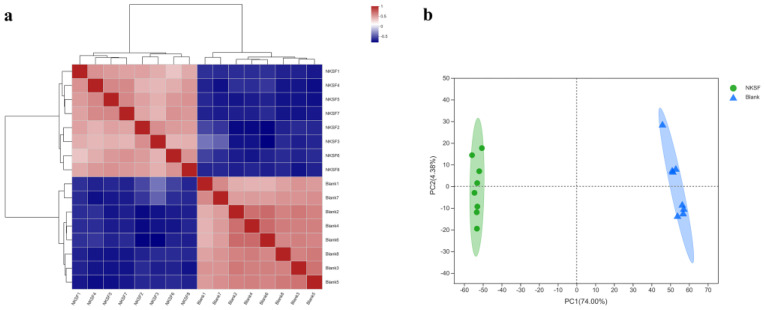
Metabolomics sample correlation analysis. (**a**) Sample correlation Heatmap; (**b**) sample PCA analysis.

**Figure 3 ijms-23-09040-f003:**
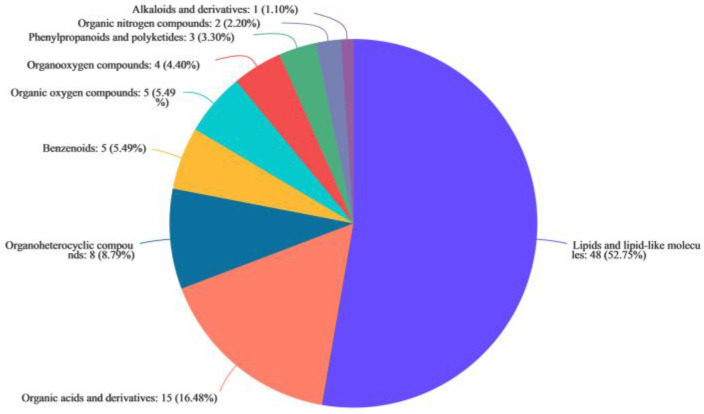
Classification of differential metabolites in the supernatant of the *Xenorhabdus bovienii* ferment.

**Figure 4 ijms-23-09040-f004:**
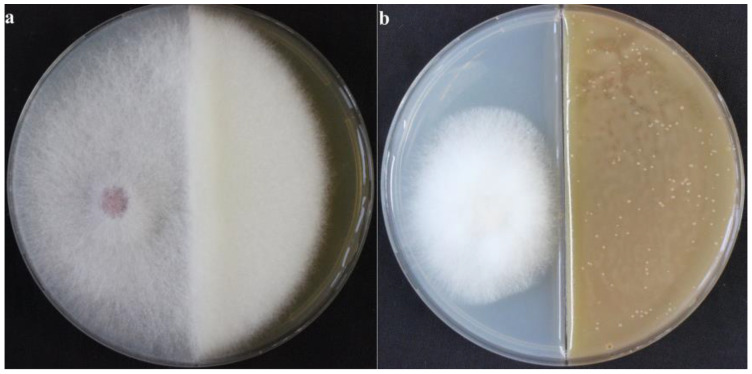
Inhibitory effect of volatile organic compounds produced by *Xenorhabdus bovienii* on NK-NH1. The plates were incubated at 28 °C for 5 days, *n* = 3. (**a**) NK-NH1 was inoculated on the left half of the plate and culture medium was applied on the right half; (**b**) NK-NH1 was inoculated on the left half of the plate and *X. bovienii* was inoculated on the right half.

**Figure 5 ijms-23-09040-f005:**
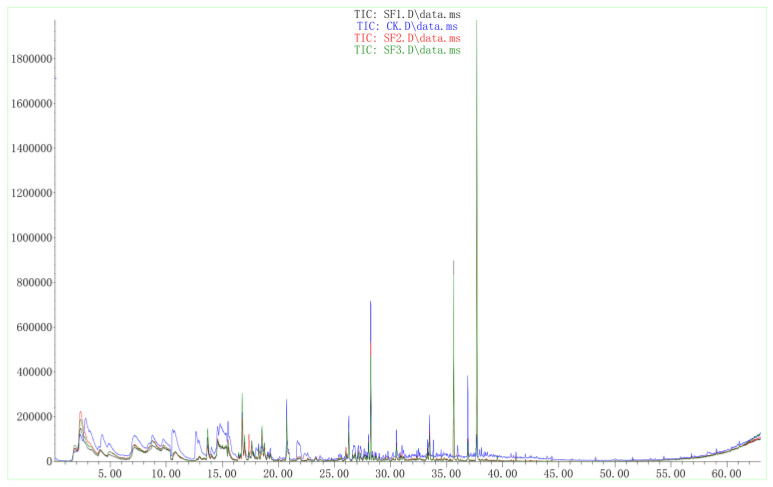
GC–MS results of volatile organic compounds produced by *Xenorhabdus bovienii*. CK is the blank control and SF1, SF2, and SF3 are three replicate samples in the experimental group.

**Figure 6 ijms-23-09040-f006:**
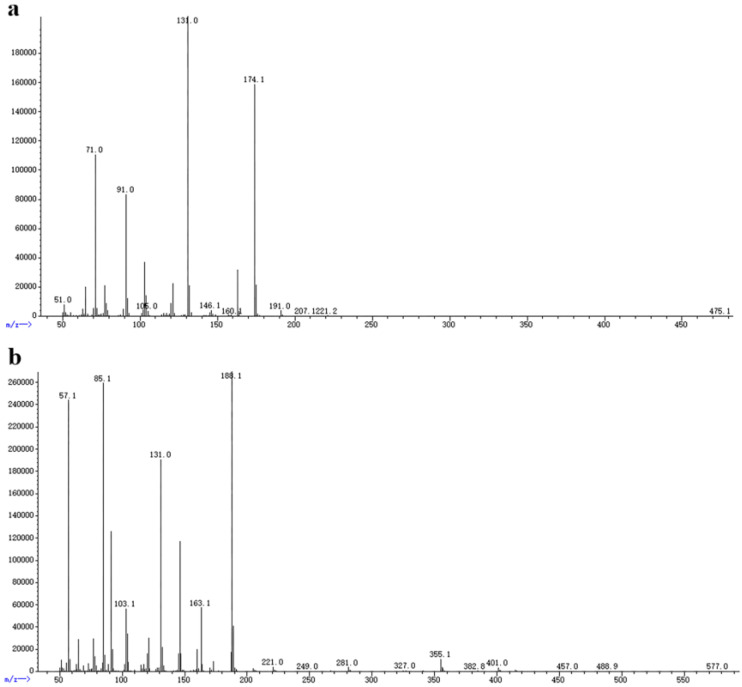
Mass spectrogram. (**a**) The retention time is 35.629 min, indicative of 2-pentanone, 3-(phenylmethyl); (**b**) the retention time is 37.685 min, indicative of 1-hexen-3-one, 5-methyl-1-phenyl.

**Figure 7 ijms-23-09040-f007:**
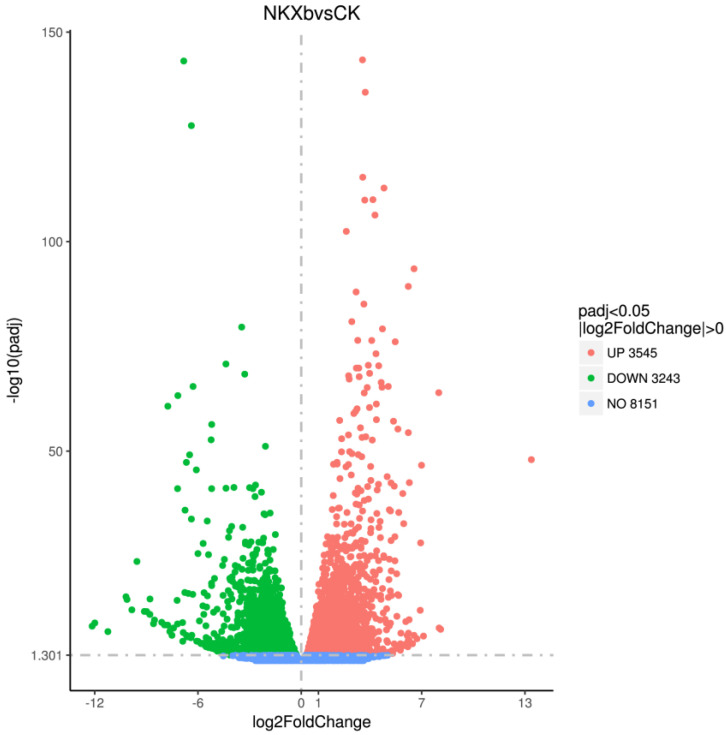
Analysis of differentially expressed genes. The abscissa represents the fold change (log2FoldChange) in the expression of the genes in experimental and control groups, and the ordinate represents the significance level (−log10padj) of the difference in expression of the genes in the two groups. The upregulated and downregulated genes are indicated by red and green dots, respectively.

**Figure 8 ijms-23-09040-f008:**
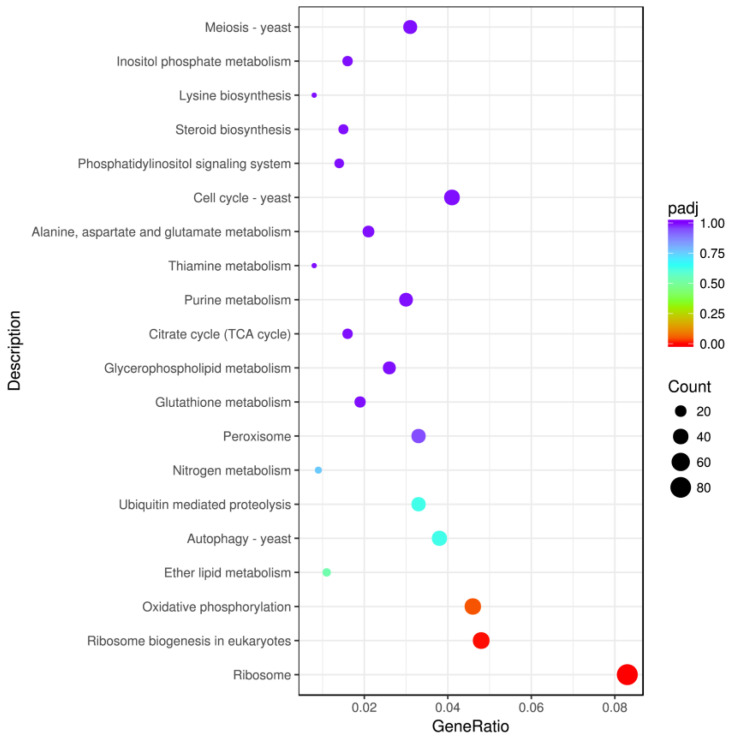
KEGG enrichment analysis. The abscissa shows the ratio of the number of differentially expressed genes annotated using the KEGG pathway to the total number of differentially expressed genes. The ordinate shows the KEGG pathways. The size of the dots represents the number of genes annotated to the KEGG pathway and the color from red to purple represents the significance of enrichment.

**Table 1 ijms-23-09040-t001:** Main differential metabolites.

Metabolite	Formula	FC (NKSF/Blank)	*p* Value
lysoPC (28:0)	C_36_H_74_NO_7_P	4643.192488	1.47 × 10^−14^
N-Carbamoylputrescine	C_5_H_13_N_3_O	565,313.1885	2.67 × 10^−11^
20-HETE ethanolamide	C_22_H_37_NO_3_	462.7804571	9.11 × 10^−11^
PE-NMe (16:0/22:5(4Z,7Z,10Z,13Z,16Z))	C_44_H_78_NO_8_P	403.5201482	3.26 × 10^−14^
alpha-Ionol O-[arabinosyl-(1->6)-glucoside]	C_24_H_40_O_10_	525.0169799	4.03 × 10^−8^
PG (16:0/22:6(4Z,7Z,10Z,13Z,16Z,19Z))	C_44_H_75_O_10_P	269.3971166	6.03 × 10^−10^
PA (16:0/18:1(11Z))	C_37_H_71_O_8_P	108.0167793	3.29 × 10^−9^
PA (16:0/15:0)	C_34_H_67_O_8_P	105.6211958	1.09 × 10^−5^
Lucidenic acid M	C_27_H_42_O_6_	25,115.625	1.36 × 10^−6^
Validamycin A	C_20_H_35_NO_13_	1875.077176	1.74 × 10^−7^
Octadecyl fumarate	C_22_H_40_O_4_	263.6363636	3.50 × 10^−15^
Lucyoside R	C_36_H_58_O_11_	196.4816356	7.89 × 10^−12^
2,6,6-Trimethyl-1,4-cyclohexadiene-1-carboxaldehyde	C_10_H_14_O	186.2414338	1.39 × 10^−9^
1-(2,3-Dihydro-1H-pyrrolizin-5-yl)-2-propen-1-one	C_10_H_11_NO	6173.232908	4.04 × 10^−10^

The bacterial fermentation cultured for 2 days was used as the experimental group (NKSF) and TSY liquid medium was used as the control group (Blank).

## Data Availability

Raw data can be found at https://www.ncbi.nlm.nih.gov/geo/query/acc.cgi?acc=GSE207994 (accessed on 27 June 2022).

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
