# Peer review of "Antifungal Substances Produced by Xenorhabdus bovienii and Its Inhibition Mechanism against Fusarium solani"

_ijms, 2022, doi:10.3390/ijms23169040_

Round 1

Reviewer 1 Report

This is an interesting study to identify novel antimicrobial compounds.

General comments:

            The introduction and discussion could be more clear as to whether this work extended from the presence of interesting organisms associated with decaying artifacts, or to identify strategies to protect artifacts in the natural world?  Either way, is identification of metabolites from organisms grown in artificial laboratory environments (controlled temp, shaker, nutrient maintained, etc) informative to those found in the natural environment?  In addition, if protective strategies are sought, how would they be used in a river environment (in depth discussion not expected, just an acknowledgment of the challenges).

            Much of the discussion revolves around bacteriostatic compounds. It is understandable if that represents much of the published work, but a clarifying sentence would help the reader.  It almost reads as if the discussion is from a different set of work regarding bacterial inhibition, rather than this paper aimed at fungal inhibition.

Specific comments:

·      Figure 1 appears to be an n=1 experiment.  Were the effects on growth quantified with sufficient replicates?  This data should be provided in main paper.  The remaining studies are based on the growth (or inhibition of growth) observed in panel C.  The colony observed in that panel looks strikingly different than the others – beyond just less growth.  This in an unconvincing piece of data and troublesome that it is the basis of the rest of the project.

·      The text associated with Figures 2, 3, 5 and 6 is illegible and unacceptable for analysis.

·      Figure 4 appears to be an n=1 experiment.  Appropriate replicates should be assessed and quantified and the data presented in the main paper.

The discussion associated with Figure 6 suggests that the identified VOC are anti-proliferative.  Were specific studies actually done to isolate and assess these compounds in a growth study?

·      Was the sheer number of differentially expressed genes surprising?  What percentage of the total number of genes in the organism does this represent?  Ribosomal genes seem to be commonly identified in differential expression studies.  Do you think that is the case here as well?

·      Genes associated with autophagy are in both the up- and downregulated pools.  Since they are highlighted in the text, some discussion should also be provided as to their relevance.

·      Referencing genes as ‘upregulated’ at >1-fold (in the discussion) is inconsistent with the standards in the field.  Perhaps better would be to indicate several genes of interest appeared to be differentially regulated, but did not meet the 2-fold standard.

Minor:  Some sentence structure revision would improve readability.

             Be consistent with the spelling of ‘artifact’ (vs artefact)

             Not sure the word ‘fermented’ needs to be included when referring to the

                   culture medium or supernatant.  It almost implies that additional

                   procedures were involved.

Author Response

This is an interesting study to identify novel antimicrobial compounds.

Response: Thank you for your comments.

Point 1: The introduction and discussion could be more clear as to whether this work extended from the presence of interesting organisms associated with decaying artifacts, or to identify strategies to protect artifacts in the natural world?  Either way, is identification of metabolites from organisms grown in artificial laboratory environments (controlled temp, shaker, nutrient maintained, etc) informative to those found in the natural environment?  In addition, if protective strategies are sought, how would they be used in a river environment (in depth discussion not expected, just an acknowledgment of the challenges).

Response 1: Thank you for your comments. In order to find more green antifungal substances to prevent and control the damage of NK-NH1 to the Nanhai No. 1 Shipwreck, we have tried to find the substances from microbes and plants. Nematode endosymbiotic bacteria have inhibitory effects on some fungal diseases of crop (Shan. S, et al. Microbial Biotechnology. 2019). We screened nematode endosymbiotic bacteria and found that it also have inhibitory effects on NK-NH1. The main content of this paper is to study the antifungal substances produced by X. bovienii and its inhibitory mechanism on NK-NH1. At present, Nanhai No. 1 Shipwreck has been salvaged and stored in the museum. It has been out of the water environment and exposed to the air. Later the applications of antifungal substances will also be implemented in the museum environment.

Point 2: Much of the discussion revolves around bacteriostatic compounds. It is understandable if that represents much of the published work, but a clarifying sentence would help the reader.  It almost reads as if the discussion is from a different set of work regarding bacterial inhibition, rather than this paper aimed at fungal inhibition.

Response 2: Thank you for your comments. This article mainly studies the antifungal substances produced by symbiotic bacteria in nematodes and the mechanism of inhibiting fungi. The antifungal substances produced by symbiotic bacteria in nematodes in liquid medium and gas were discussed (The first and second paragraphs of the discussion).

Point 3: Figure 1 appears to be an n=1 experiment.  Were the effects on growth quantified with sufficient replicates?  This data should be provided in main paper.  The remaining studies are based on the growth (or inhibition of growth) observed in panel C.  The colony observed in that panel looks strikingly different than the others – beyond just less growth.  This in an unconvincing piece of data and troublesome that it is the basis of the rest of the project.

Response 3: Thank you for your comments. Figure 1 is not an n=1 experiment. In order to select the fermentation supernatant with the best antifungal effect for metabonomic analysis, we conducted three repeated experiments. The experimental results only selected one group of repetition group. In addition, the colony in Fig. 1C is determined to be the same as those in other figures, only because the colony morphology of NK-NH1 has changed after being inhibited. The metabolites in the fermentation supernatant inhibited the growth and sporulation of NK-NH1.

Point 4: The text associated with Figures 2, 3, 5 and 6 is illegible and unacceptable for analysis.

Response 4: Thank you for your comments. We have compressed the images in the manuscript for easier manuscript upload, and we have also submitted high-resolution images at the same time when we submit the manuscript. Now we have directly added the high-resolution pictures to the manuscript.

Point 5: Figure 4 appears to be an n=1 experiment. Appropriate replicates should be assessed and quantified and the data presented in the main paper.

 Response 5: Thank you for your comments. Figure 4 is not an n=1 experiment. In order to detect the changes in gene expression after NK-NH1 was inhibited by VOCs produced by X. bovienii, we conducted three repeated experiments. The experimental results only selected one group of repetition group. In addition, we have three parallel samples for transcriptomics, which also shows that this is not an n=1 experiment.

Point 6: The discussion associated with Figure 6 suggests that the identified VOC are anti-proliferative. Were specific studies actually done to isolate and assess these compounds in a growth study?

Response 6: Thank you for your comments. No specific work in this area has yet been done. We will consider isolating and assessing these compounds in our subsequent experiments.

 Point 7: Was the sheer number of differentially expressed genes surprising?  What percentage of the total number of genes in the organism does this represent?  Ribosomal genes seem to be commonly identified in differential expression studies.  Do you think that is the case here as well?

 Response 7:  Thank you for your comments. Genes with an adjusted P-value <0.05 found by DESeq2 were assigned as differentially expressed.  According to this standard, we found 6788 differentially expressed genes, accounting for 39.0% of the total number of genes. The significant enrichment is mainly related to ribosome pathway (ribosome pathway, ribosome biosynthesis pathway of eukaryotes), but it does not mean that the differential genes in the pathway are ribosome genes, only related to ribosome synthesis. KEGG enrichment results are analyzed by comparing the differential genes in the KEGG database. Ribosome related pathways are only significantly enriched in this result, which does not mean that other pathways have no research significance, and the genes in the pathway may also participate in other metabolic pathways

Point 8: Genes associated with autophagy are in both the up- and downregulated pools. Since they are highlighted in the text, some discussion should also be provided as to their relevance.

Response 8:  Thank you for your comments. We discussed the relationship between genes in autophagy pathway and some mechanisms (the third paragraph of the discussion).

Point 9: Referencing genes as ‘upregulated’ at >1-fold (in the discussion) is inconsistent with the standards in the field.  Perhaps better would be to indicate several genes of interest appeared to be differentially regulated, but did not meet the 2-fold standard.

 Response 9: Thank you for your reminder. We have changed the relevant description.

Point 10: Some sentence structure revision would improve readability.

Response 10: Thank you for your comments. We have turned to a native English speaker to improve the language.

Point 11: Be consistent with the spelling of ‘artifact’ (vs artefact)

 Response 11: Thank you for your comments. We have changed all ‘artefact‘ to ‘artifact’.

Point 12: Not sure the word ‘fermented’ needs to be included when referring to the culture medium or supernatant. It almost implies that additional procedures were involved.

Response 12: Thank you for your comments. We have changed it to the supernatant

Reviewer 2 Report

According to the manuscript, Antifungal substances produced by Xenorhabdus bovienii and its inhibition mechanism against Fusarium solani. This is an interesting study that looked at the inhibitory mechanisms of nematode endosymbiotic bacteria used to inhibit Fusarium solani NK-NH1, the surface disease fungus that dominates the hulls of the Nanhai No. 1 Shipwreck. This manuscript is recommended for publication after minor revision. Please find the reviewer comments below.

1.     Please turn to a native English speaker to improve the language.

2.   In Figure 1, why does the fermentation of X. bovienii supernatant at 2 days incubation time show the best inhibitory effect? Why the longer incubation time does not show an inhibition effect (3, 4, 5 days)?

3.     All picture data is unclear. Please check the image resolution and adhere to the journal format.

4.     The authors should indicate the compound name on the result data in Figure 6. 

Author Response

According to the manuscript, Antifungal substances produced by Xenorhabdus bovienii and its inhibition mechanism against Fusarium solani. This is an interesting study that looked at the inhibitory mechanisms of nematode endosymbiotic bacteria used to inhibit Fusarium solani NK-NH1, the surface disease fungus that dominates the hulls of the Nanhai No. 1 Shipwreck. This manuscript is recommended for publication after minor revision. Please find the reviewer comments below.

 Response: Thank you for your comments. We have revised the article according to your suggestions.

Point 1: Please turn to a native English speaker to improve the language.

Response 1: Thank you for your comments. We have turned to a native English speaker to improve the language.

Point 2: In Figure 1, why does the fermentation of X. bovienii supernatant at 2 days incubation time show the best inhibitory effect? Why the longer incubation time does not show an inhibition effect (3, 4, 5 days)?

Response 2:  Thank you for your comments. The metabolic process of X. bovienii is very complex. The metabolites with antifungal activity may be intermediate metabolites, which are produced most at 2 days incubation time, and then consumed by other metabolic processes.

Point 3: All picture data is unclear. Please check the image resolution and adhere to the journal format.

Response 3: Thank you for your comments. We have compressed the images in the manuscript for easier manuscript upload, and we have also submitted high-resolution images at the same time when we submit the manuscript. Now we have directly added the high-resolution pictures to the manuscript.

Point 4: The authors should indicate the compound name on the result data in Figure 6.

Response 4:  Thank you for your comments. We indicated the name of the compounds in the note of Figure 6.